# Characterization of Flexible Amorphous Silicon Thin-Film Transistor-Based Detectors with Positive-Intrinsic-Negative Diode in Radiography

**DOI:** 10.3390/diagnostics12092103

**Published:** 2022-08-30

**Authors:** Bongju Han, Minji Park, Kyuseok Kim, Youngjin Lee

**Affiliations:** 1Quality Assurance Team, Business Division, Vieworks, 41-3, Burim-ro 170beon-gil, Dongan-gu, Anyang-si 14055, Korea; 2Department of Radiological Science, College of Health Science, Gachon University, 191, Hambakmoe-ro, Yeonsu-gu, Incheon 21936, Korea; 3Department of Health Science, General Graduate School of Gachon University, 191, Hambakmoe-ro, Yeonsu-gu, Incheon 21936, Korea; 4Department of Integrative Medicine, Major in Digital Healthcare, Yonsei University College of Medicine, Unju-ro, Gangman-gu, Seoul 06229, Korea

**Keywords:** digital radiography, thin-film transistor detector, flexible amorphous silicon, performance evaluation of an X-ray imaging system

## Abstract

Low-dose exposure and work convenience are required for mobile X-ray systems during the COVID-19 pandemic. We investigated a novel X-ray detector (FXRD-4343FAW, VIEWORKS, Anyang, Korea) composed of a thin-film transistor based on amorphous silicon with a flexible plastic substrate. This detector is composed of a thallium-doped cesium iodide scintillator with a pixel size of 99 μm, pixel matrix of 4316 × 4316, and weight of 2.95 kg. The proposed detector has the advantages of high-noise characteristics and low weight, which provide patients and workers with an advantage in terms of the dose and work efficiency, respectively. We performed a quantitative evaluation and an experiment to demonstrate its viability. The modulation transfer function, noise power spectrum, and detective quantum efficiency were identified using the proposed and comparative detectors, according to the International Electrotechnical Commission protocol. Additionally, the contrast-to-noise ratio and coefficient of variation were investigated using a human-like phantom. Our results indicate that the proposed detector efficiently increases the image performance in terms of noise characteristics. The detailed performance evaluation demonstrated that the outcomes of the use of the proposed detector confirmed the viability of mobile X-ray devices that require low doses. Consequently, the novel FXRD-4343FAW X-ray detector is expected to improve the image quality and work convenience in extended radiography.

## 1. Introduction

Digital radiography (DR) has gained acceptance in the medical and nondestructive testing fields owing to its workflow advantages [1]. In the 1970s, film/screen analog X-ray imaging systems accounted for the majority of radiographic devices owing to their superior spatial resolution and low cost. However, the film/screen method has limitations, such as film storage location, environmental pollution in the development process, integrated management of image data, and utilization of existing data [2]. To overcome these problems, researchers have developed digital systems for projection radiography, which can be divided into two broadly defined categories: computed tomography (CR) and DR [3,4]. The clinical application of the CR system was introduced by Fuji in the early 1980s. It uses a phosphor storage plate for radiation detection, which is composed of photostimulable phosphor crystals (e.g., BaFX:Eu^2+^, where X is a halogen). When the phosphor storage plate is exposed to radiation, the photostimulable phosphor crystals enter an excited state and then irradiate the laser at the detection layer. At this moment, the stored energy is emitted as visible light, and the photodiode converts it into an electric signal according to the amount of light [5]. This approach has a superior dynamic range over the film/screen method and can reduce the patient dose [3]. However, this requires an additional device to read the information, and real-time radiography cannot be performed.

DR has been suggested as an alternative in solving the problems associated with existing film/screen methods and CR. The DR method is divided into direct and indirect conversions according to the method of radiation detection [6]. The direct conversion method converts a photon to an electric signal using a photoconductor (e.g., amorphous selenium [a-Se], thallium bromide, and gadolinium compounds) [7]. Therefore, this approach has the merit of extremely high intrinsic spatial resolution and mainly uses mammography [8,9]. In contrast, the indirect conversion method uses scintillators (e.g., thallium-doped cesium iodide (CsI[Tl]), gadolinium oxisulfide [Gd_2_O_2_S:Tb], barium fluoride, and cadmium tungstate) to improve the conversion rate between incident photons and electronic charges [10,11]. Typically, a-Se has a lower mass attenuation coefficient than CsI in the energy range above 40 keV [12]. After conversion of photons into light, they are converted into electrical signals using a coupled charge device and complementary metal-oxide-semiconductor (CMOS). Recently, large-area hydrogenated amorphous silicon (a-Si) thin-film transistor (TFT)-based flat-panel detectors have been widely used in common radiography.

The DR system has demands of high image performance under low radiation exposure conditions to protect patients and workers [13,14]. The a-Si TFT-based flat-panel detector, which is currently the most widely used device, has an important limitation in that it has relatively high electronic noise characteristics under low-exposure conditions [13,14,15]. To overcome this problem, researchers have applied various methods to signal processing, including the reduction of additional electronic noise and optimization of the system gain [14,16]. Another approach is to increase the scintillator thickness to increase the number of light photons. However, this technique results in resolution degradation and an increased weight. Work convenience issues were prompted by COVID-19 as a momentum. The process of wearing a protective suit while performing roentgenographic procedures is a significant burden on the operator.

In this study, we examined a novel indirect conversion detector with a flexible a-Si TFT and a positive-intrinsic-negative diode. It exhibits better noise characteristics by adjusting the optimal scintillator thickness while preserving the spatial resolution. In addition, it reduces the weight by using a flexible TFT substrate instead of a conventional glass substrate. This has the advantage of improving the work efficiency of mobile X-ray devices. In the following sections, we present the experimental results and discuss the performance of the proposed detector and existing indirect conversion detectors. Quantitative evaluation was conducted by calculating the modulation transfer function (MTF), normalized noise power spectrum (NNPS), and detective quantum efficiency (DQE), according to the International Electrotechnical Commission (IEC) protocol. Additionally, the contrast-to-noise ratio (CNR) and coefficient of variation (COV) were evaluated using a human-like phantom.

## 2. Materials and Methods

### 2.1. X-ray Detectors

Three DR detectors (VIEWORKS, Anyang, Korea) were used in the experiment. Two DR detectors (FXRD-4343VAW and FXRD-4343VAW Plus, VIEWORKS) were used for comparison with the proposed detector in terms of performance. All detectors consisted of a CsI(Tl) scintillator and a-Si TFT. The thickness of the scintillator increased in the following order: FXRD-4343VAW, FXRD-4343VAW Plus, and FXRD-4343FAW. FXRD-4343VAW and FXRD-4343VAW Plus had a pixel spacing of 140 μm and pixel matrix of 3072 × 3072. The proposed detector had a high pixel resolution (pixel spacing = 99 µm, pixel matrix = 4316 × 4316) because it compensates for light blurring in relation to the thick scintillator. The two detectors used conventional glass-type substrates in the TFT in contrast to the proposed detector-designed plastic substrates, which are mainly used in flexible TFTs. Plastic substrate-based TFTs are much lighter than glass-based TFTs. Despite the thick scintillator, the weight of the proposed detector (FXRD-4343FAW) was approximately 1.16 times lower than that of the FXRD-4343VAW detector. The detailed specifications of the detectors are listed in Table 1.

### 2.2. Experimental Setup

The image quality of X-ray systems is generally influenced by the radiation exposure condition (e.g., kV, mA, or exposure time). Thus, the detector performance was compared under similar X-ray beam conditions, and three detectors were evaluated in accordance with the IEC standard for general radiography (IEC 62220-1-1:2015 International Standard) [17,18]. IEC 62220-1-1:2015 has updated the Radiation Quality (RQA)-3 and RQA-5 compared with IEC 62220-1:2003; therein, the half-value layer thickness is 3.8 and 6.8 mm, instead of 4 and 7.1 mm, respectively, and the squared signal-to-noise ratio per air kerma is 21,759 and 20,673 (1/mm^2^ × μGy), instead of 30,174 and 29,653 (1/mm^2^ × μGy), respectively. In addition, IEC 62220-1-1:2015 has provided the method for the determination of the lag effects [19]. Figure 1 shows the conditions in the RQA-5 at 70 kV and additional filter at 21.0 mmAl. Herein, the X-ray tube used (E7252X, Toshiba, Kabushikigaisha, Japan) had a focal spot size of 1.2/0.6 mm, a Tungsten target, and an anode angle of 16°.

### 2.3. Image Performance

Generally, the representative factors that determine the image quality are contrast, sharpness, and noise. In particular, it is important to irradiate patients at a dose as low as possible, except for diagnostics [20]. Therefore, investigation of the image performance of radiologic equipment is useful for improving the diagnostic accuracy. The IEC protocol proposes the use of the MTF, NNPS, and DQE to uniformly express the representative factors according to the spatial frequency.

The MTF is a mathematical function that shows the contrast reduction of the spatial frequency through the imaging system. The classic methods for measuring the MTF can be divided into three approaches: pinhole [21,22], slit [23], and edge phantom [24] methods. All these methods aim to calculate the point spread function (PSF), which are generally defined in Equation (1):(1)$$\mathrm{PSF}\left(x,y\right)=\frac{1}{\sigma \sqrt{2\pi }}\exp\left(-\frac{x^{2}+{\;y}^{2}}{2\sigma^{2}}\right),$$
where x and y denote the orthogonal coordinate indices, and σ is the standard deviation, which assumes the form of a Gaussian distribution and the linear and shift invariant in the imaging system [25]. Herein, the pinhole and slit methods require ϕ of 10 μm, which is difficult to manufacture. Therefore, IEC 62220-1-1:2015 has provided a protocol using an edge phantom (e.g., 1-mm tungsten [W] foil)-based technique; herein, we also conducted the experiment in compliance with the protocol. The line spread function (LSF) was calculated to differentiate the edge spread function, which can be acquired by tilting the edge phantom to avoid potential aliasing [26]. The MTF is represented as follows:(2)MTF(f)=ℱ{LSF},
where f is a one-dimensional coordinate in the frequency domain, and ℱ is an indicator of the Fourier transformation. Thus, it is expressed as a value of 1 or less when resolution loss occurs.

The NNPS is also a mathematical function of noise variance in the spatial frequency. Ten white images obtained according to the RQA-5 and homogeneous regions (1024 × 1024 pixels) were selected. The 2D NNPS was derived using the Fourier transformation of 128 × 128 or 256 × 256 half-overlapping regions of interest (ROIs), which were normalized and subtracted from 1024 × 1024 images [27]; it can be calculated using Equation (3):(3)$$\mathrm{NNPS}\left(u,v\right)=\frac{\sum_{m=1}^{M}\sum_{l=1}^{L}{\left|ℱ\left\{I\left(x_{l},y_{m}\right)-S\left(x_{l},y_{m}\right)\right\}\right|}^{2}}{{\left(\;\bar{S}\left(x,y\right)\right)}^{2}},$$
where u and v denote the coordinate indices in the frequency domain; $I\left(x_{l},y_{m}\right)$ and $S\left(x_{l},y_{m}\right)$ are acquired, as well as the average ROI of l^th^ and m^th^, respectively. $\;\bar{S}\left(x,y\right)$ is the average signal of 10 white images. Finally, the 1D NNPS represented the radial and axial averages using seven frequency bins [28]. A small NNPS value according to the spatial frequency indicated the superior noise characteristics of the imaging system.

The DQE shows the ability to transfer information from the system input to the output. It can be calculated using the MTF, NNPS, and number of photons (Φ), as follows [29]:(4)$$\mathrm{DQE}\left(f\right)=\frac{{\mathrm{SNR}}_{\mathrm{out}}^{2}}{{\mathrm{SNR}}_{\mathrm{in}}^{2}}=\frac{{\mathrm{MTF}}^{2}\left(f\right)}{\Phi \times {\mathrm{NNPS}}^{2}\left(f\right)}=\frac{{\mathrm{MTF}}^{2}\left(f\right)}{\frac{\Phi }{K_{a}}\times K_{a}\times {\mathrm{NNPS}}^{2}\left(f\right)},$$
where ${\mathrm{SNR}}_{\mathrm{in}}^{2}$ indicates the Φ of the input quanta (photon fluence per unit area (mm^2^)). For the convenience of measurement, Φ was substituted for the photon fluence per air kerma ratio ($K_{a}$, photon fluence per mm^2^ per μGy). The DQE was between 0 and 1, and a higher DQE value indicated a highly efficient imaging system.

To demonstrate the viability in the clinical images, we evaluated the CNR [30] and COV [31]. The CNR and COV are generally calculated as follows:(5)$$\mathrm{CNR}=\frac{\left|S_{{\mathrm{ROI}}_{A}}-S_{{\mathrm{ROI}}_{B}}\right|}{\sqrt{\sigma_{{\mathrm{ROI}}_{A}}{}^{2}+\sigma_{{\mathrm{ROI}}_{B}}{}^{2}}},$$
(6)$$\mathrm{COV}=\frac{\sigma_{\mathrm{ROI}}}{S_{\mathrm{ROI}}},$$
where $S_{{\mathrm{ROI}}_{A}}$ and $\sigma_{{\mathrm{ROI}}_{A}}$ are the mean and standard deviation of the intensity in the ROI, respectively; $S_{{\mathrm{ROI}}_{B}}$ and $\sigma_{{\mathrm{ROI}}_{B}}$ are the mean and standard deviation of the background intensity in the ROI, respectively.

A normal workstation (OS: Windows 10, CPU: AMD Ryzen 7 3700X, RAM: 256 GB) was used to evaluate the MTF, NNPS, DQE, CNR, and COV. The MTF, NNPS, and DQE were investigated using free open software [32], and the CNR and COV were calculated using a handmade code based on MATLAB (R2021b, Mathworks, CA, USA).

## 3. Results and Discussion

Figure 2 shows the plots of the (a) MTF, (b) NNPS, and (c) DQE using the three detectors.

The MTF results showed that the MTF value of the FXRD-4343VAW detector at all spatial frequencies was higher than that of the FXRD-4343VAW Plus and FXRD-4343FAW detectors. The MTF value of the proposed FXRD-4343FAW detector slightly increased at 1.5 lp/mm compared to that of the FXRD-4343VAW Plus detector. The pixel size of the proposed detector was 0.7-fold smaller than that of the FXRD-4343VAW and FXRD-4343VAW Plus detectors. However, the scintillator of the FXRD-4343FAW detector was thicker than that of the two detectors, which affected the MTF degradation. In contrast, the NNPS graph of the FXRD-4343FAW detector showed the lowest value (approximately $5\times 10^{-7}{\mathrm{mm}}^{2}$ at a 3.5-lp/mm spatial frequency), which was approximately 16 and 5 times lower than that of the FXRD-4343VAW and FXRD-4343VAW Plus detectors, respectively. Contrary to the MTF results, this NNPS result indicated that the scintillator of the proposed FXRD-4343FAW detector was thicker than that of the other detectors, which reduced the noise variation. Although the pixel size of the FXRD-4343FAW detector was smaller than that of the other detectors, the better NNPS performance indicated that the proposed detector was optimized. Finally, the DQE plot of the proposed FXRD-4343FAW detector was almost higher than that of the other detectors at all spatial frequencies. As a result, the proposed FXRD-4343FAW detector was highly efficient, sensitive to noise, and proven to be useful in conventional radiographic conditions.

Among the factors that can be used to evaluate the performance of an X-ray detector, spatial resolution and noise are representative factors that determine the image quality. The spatial resolution is evaluated using the MTF as a factor that indicates the extent to which a desired area can be distinguished in an X-ray image. The MTF graph is dependent on the shape of the X-ray detector and has the characteristic that the value varies depending on the measurement method. Rivertti et al. compared the MTF characteristics of detectors using gadolinium-oxysulfide phosphor and CsI(Tl) and confirmed that the tendency changes depending on the type of detector material [33]. In addition, by comparing the MTF data obtained under various RQA conditions, they confirmed a slight difference [33]. Based on the results of the RQA-5 condition used in this study, it is necessary to analyze the results under other conditions, depending on the future measurement situation.

The spatial resolution of the image, noise, and sensitivity characteristics also change depending on the thickness of the X-ray detector. In general, as the thickness of the detector increases, the spatial resolution deteriorates; however, the noise and sensitivity characteristics improve with an increase in the signal amount. Miller et al. confirmed that the curve of the MTF graph changed rapidly as the thickness of the CsI(Tl)-based X-ray detector increased [34]. In addition, Zambon et al. demonstrated that the thicker the detector, the higher the DQE value under various RQA conditions [35]. The results of these preliminary studies showed the same trend as the results of this study; if the proposed X-ray detector with a relatively thick size is used in the medical field, it is expected that the image quality characteristics and patient exposure to radiation will be reduced. Based on the results of this study, it is expected that future research on the optimization of the MTF, NNPS, and DQE values according to the thickness and shape of the detector will be conducted to suggest guidelines for using the optimal detector for each field.

Figure 3 shows radiographic images of the human-like phantom at the positions of the foot and pelvis. These images were obtained in the anteroposterior (AP) position. The foot AP images were obtained using the three detectors in each exposure condition to achieve the best image quality: FXRD-4343VAW: 50 kV_p_, 2.8 mAs, and dose area product (DAP) = 42.41 μGy; FXRD-4343VAW Plus: 50 kV_p_, 2.0 mAs, and DAP = 28.71 μGy; and FXRD-4343FAW: 50 kV_p_, 2.0 mAs, and DAP = 28.71 μGy. The pelvis AP images were also obtained under the following conditions: FXRD-4343VAW: 75 kV_p_, 8.0 mAs, and DAP = 438.4 μGy; FXRD-4343VAW Plus: 75 kV_p_, 5.6 mAs, and DAP = 305.8 μGy; and FXRD-4343FAW: 75 kV_p_, 5.6 mAs, and DAP = 305.8 μGy.

Figure 4 shows the CNR and COV with the ROI shown in Figure 3 for the quantitative comparison. The evaluated CNRs for the foot AP images obtained using the FXRD-4343VAW, FXRD-4343VAW Plus, and FXRD-4343FAW detectors were 84.5501, 90.2716, and 96.7472, respectively. Meanwhile, the evaluated CNRs for the pelvis AP images obtained using the FXRD-4343VAW, FXRD-4343VAW Plus, and FXRD-4343FAW detectors were 63.3110, 69.4594, and 105.0559, respectively. In the comparison of the tendencies of the CNR of the two human-like phantom images, it was confirmed that the contrast and signal of the proposed FXRD-4343FAW detector were higher than those of the other detectors. The calculated COV for the foot AP image obtained using the FXRD-4343VAW, FXRD-4343VAW Plus, and FXRD-4343FAW detectors, which evaluated ROI_A_ in Figure 3, was 0.0107, 0.0089, and 0.0057, respectively. Meanwhile, the calculated COV for the pelvis AP images obtained using the three detectors was 0.0074, 0.0065, and 0.0010, respectively. Taken together, these results demonstrate that the proposed detector has superior image performance with regard to the number of signal and noise characteristics.

Noise inevitably generated in medical X-ray images is one of the greatest factors that reduce the accuracy of lesion diagnosis. As mentioned earlier, the amount and distribution of noise vary depending on the thickness and shape of the X-ray detector, as shown in Figure 4. We proved that noise can be reduced using the system proposed in this study. As shown in Figure 4a, the CNR improved by 1.14 and 1.07 times when the proposed detector was used for the foot AP image compared with that when the FXRD-4343VAW and FXRD-4343VAW Plus detectors were used, respectively. In addition, we proved that the COV for the foot AP image improved by 1.88 and 1.56 times when the proposed detector was used compared with those when the FXRD-4343VAW and FXRD-4343VAW Plus detectors were used, respectively. As shown in Figure 4b, the CNRs for the pelvis AP image improved by 1.56 and 1.51 times when the proposed detector was used compared with those when the FXRD-4343VAW and FXRD-4343VAW Plus detectors were used, respectively. In addition, the COV for the pelvis AP image improved by 7.40 and 6.50 times when the proposed detector was used compared with those when the FXRD-4343VAW and FXRD-4343VAW Plus detectors were used, respectively. Image quality improvement through noise reduction in the medical field using this proposed detector system is closely related to the patient’s exposure dose. In the use of X-rays in the diagnostic medical field, the “As Low As Reasonably Achievable” principle should always be considered, and it is important to reduce the dose as much as possible. In terms of the X-ray hardware, a study showed that the calculated effective dose was reduced by 52% when an additional filter was used to reduce the patient’s exposure dose [36]. However, when the dose is reduced using an additional filter, the number of photons reaching the detector is significantly reduced, which amplifies the noise in the final image. Thus, with the application of the detector system proven from the results of this study to the medical field, it will be possible to reduce the exposure dose and noise simultaneously. In addition, it is expected that the field of application can be expanded depending on the necessary situation in whole-body X-ray examination, including the foot and pelvis.

The proposed FXRD-4343FAW detector demonstrated improved noise characteristics while reducing the device weight. This is because the thickness of the scintillator increased to a higher amount of photon-to-light generation, and the plastic substrate was light compared with the conventional glass substrate. However, as mentioned above, the light spreading phenomenon increases with the thickness of the scintillator. Thus, this imaging system has a poorer spatial resolution [37,38]. As one of the methods to overcome this problem, a deconvolution method that removes blurring using software is often suggested as a supplement [39]. Conventionally, the degradation of radiographic images (h) is defined as follows.
(7)h=PSF⊗⊗f+n,
where f is the restored radiographic image obtained by deconvolving the PSF component, which depends on the shape according to the thickness of the scintillator, focal spot size, pixel size, and magnification; n is a noise component that includes quantum and electronic noises. The PSF of the system can be obtained by determining the MTF. When calculating the accurate MTF, the technique outperforms in improving the resolution of the degraded image. However, a problem may occur when the noise component also increases [40]. The deconvolution method is similar to high-pass filtering. Therefore, noise amplification is inevitable. A deep learning-based deconvolution method that considers the noise level has been suggested as an alternative method to overcome this problem [41,42]. It is difficult to perform deconvolution when the PSF is unknown (blind deconvolution). Mobile X-ray systems are more likely to require blind deconvolution than stationary X-ray systems. We previously proposed a deep learning-based MTF prediction protocol as a relational study and evaluated its performance [25]. Figure 5 shows the results of the degraded image with noise and blurring at 49 kV_p_ and 1.6 mAs and the deblurred image using the Kim et al. method [25]. As mentioned in Equation (7), the wiener2 MATLAB function was adapted in the degraded image to remove the noise component. Note that, enlarged image indicates that the image restoration including the denoising and the deconvolution is expected to improve the image quality by restoring the degraded resolution of the proposed detector. In addition, as a consecutive research and development measure, a flexible X-ray detector can be applied to improve the image quality.

As mentioned earlier, the recent research trend in X-ray detectors based on TFT arrays is to apply a form with flexible characteristics to various fields. In this study, the characteristics of a system model for medical imaging were evaluated, and various detector characteristics were analyzed using a phantom that can simulate the human body. The proposed FXRD-4343FAW detector system with proven good detector characteristics is expected to be useful for not only medical imaging but also airport security screening and non-destructive fields. Recently, in the USA and Europe, it has been recommended not to use ionizing radiation, including X-rays, for airport security screening. When the use of an X-ray-based detection system is unavoidable, a detector with high characteristics, including the best possible DQE, should be used. In addition, even in a non-destructive field, when a detector system with high X-ray detection efficiency is used, defects inside the object can be identified more accurately. In particular, we expect that the invisible part, depending on the shape of the object, can be supplemented by using the FXRD-4343FAW detector proposed by our research team.

## 4. Conclusions

The aim of this study was to confirm the image performance of an a-Si TFT-based detector with a flexible plastic substrate. The experiments demonstrated the viability of the proposed detector. The proposed FXRD-4343FAW detector shows the DQE plot at all spatial frequencies, compared to that of FXRD-4343VAW and FXRD-4343VAW Plus detectors. The NNPS plot of FXRD-4343FAW detector indicates the lowest value at 3.5 lp/mm, approximately. Here, the NNPS value of proposed detector was approximately 16 and 5 times lower than that of FXRD-4343VAW and FXRD-4343VAW Plus detector. In addition, the CNR and COV of the proposed detector indicated improved image quality roughly at 1.41 and 4.26 on average, respectively. This indicates that the noise characteristic of the proposed detector has improved performance compared to conventional detectors. Thus, the improved DQE of proposed detector was greatly influenced by the noise improvement results while increasing the scintillation thickness. Nevertheless, the proposed detector may yield unsatisfactory spatial resolution results. The MTF value of the FXRD-4343FAW detector at 1.5 lp/mm was approximately 0.4, which is about 1.25 times lower than that of the FXRD-4343VAW detector. This may inevitably occur when the scintillation thickness is increased to improve the noise characteristics under low radiation exposure conditions. This problem can be overcome by continued research and development.

## Figures and Tables

**Figure 1 diagnostics-12-02103-f001:**
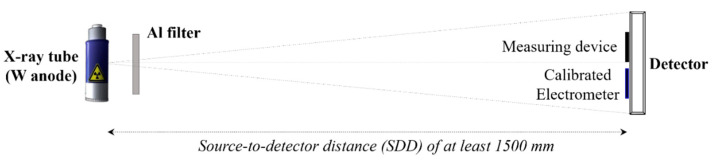
X-ray system for the experimental study, which mainly consists of an X-ray tube (E7252X, Toshiba, Kabushikigaisha, Japan), and the detectors (FXRD-4343VAW, FXRD-4343VAW Plus, and FXRD-4343FAW, VIEWORKS, Anyang, Korea). All radiation exposure conditions were set according to IEC 62220-1-1.

**Figure 2 diagnostics-12-02103-f002:**
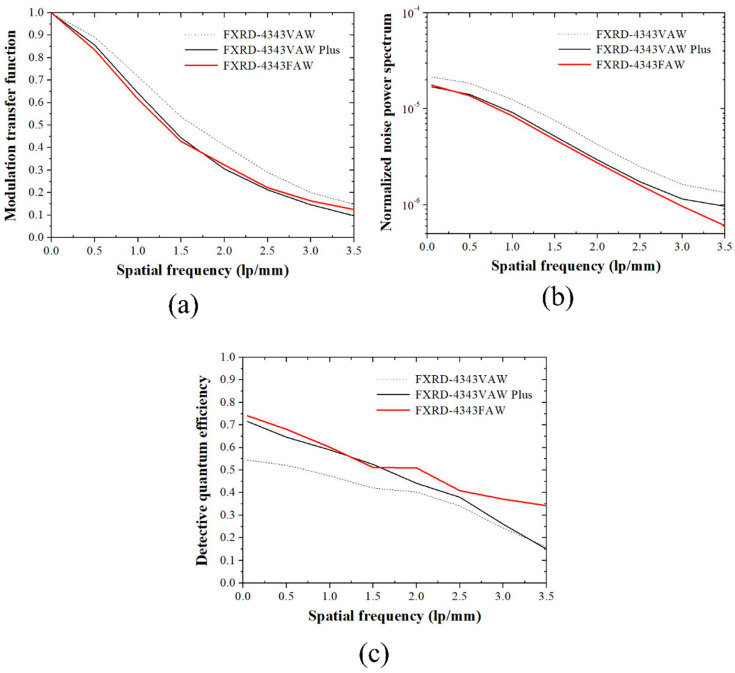
(**a**) MTF, (**b**) NNPS, and (**c**) DQE plots of the FXRD-4343VAW, FXRD-4343VAW Plus, and FXRD-4343FAW detectors according to IEC 62220-1-1:2015.

**Figure 3 diagnostics-12-02103-f003:**
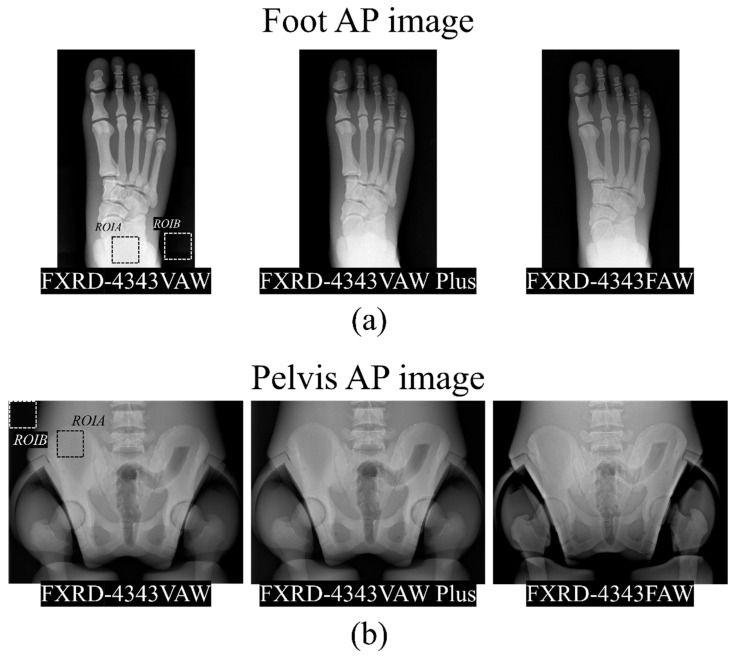
Example of (**a**) foot AP and (**b**) pelvis AP radiographic images obtained using the FXRD-4343VAW (**left**), FXRD-4343VAW Plus (**middle**), and FXRD-4343FAW (**right**) detectors.

**Figure 4 diagnostics-12-02103-f004:**
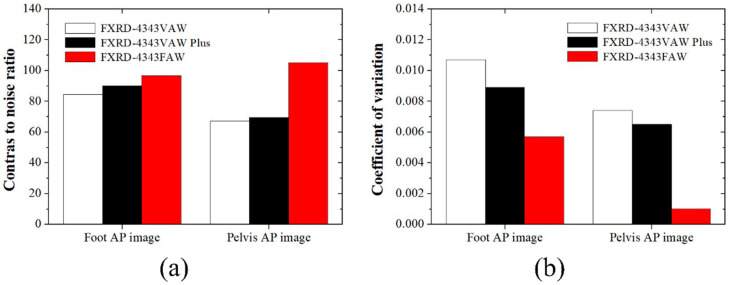
(**a**) CNR and (**b**) COV obtained using the three detectors in relation to the ROI shown in Figure 3.

**Figure 5 diagnostics-12-02103-f005:**
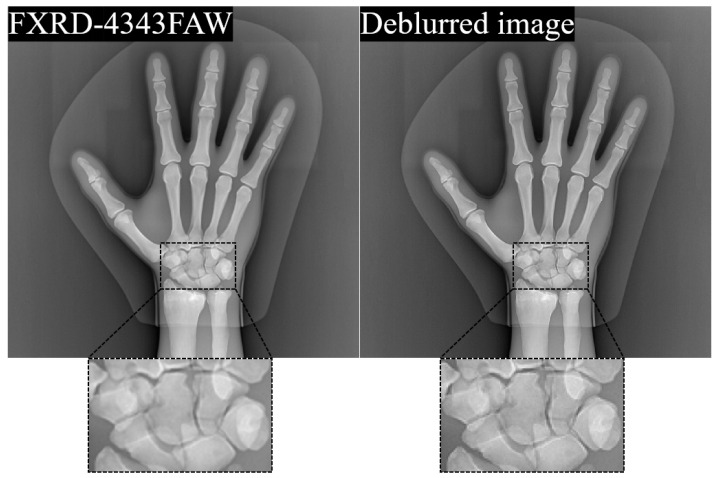
Results and enlarged images of the degraded image (**left**), which is obtained by FXRD-4343FAW detector, and its deblurred image (**right**).

**Table 1 diagnostics-12-02103-t001:** Detector specifications for the experiment.

Name	Scintillator (Thickness, vs. FXRD-4343VAW)	Pixel Size (μm)	Pixel Matrix (Pixels)	Detection Area (Pixels)	Weight (kg)
FXRD-4343VAW	CsI(Tl) (standard)	140	3072 × 3072	3048 × 3048	3.45
FXRD-4343VAW Plus	CsI(Tl) (1.5 times)	140	3072 × 3072	3048 × 3048	3.7
FXRD-4343FAW	CsI(Tl) (1.6 times)	99	4316 × 4316	4276 × 4276	2.95

## Data Availability

Not applicable.

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
