# Peer review of "Characterization of Flexible Amorphous Silicon Thin-Film Transistor-Based Detectors with Positive-Intrinsic-Negative Diode in Radiography"

_diagnostics, 2022, doi:10.3390/diagnostics12092103_

Round 1

Reviewer 1 Report

The authors investigated a novel X-ray detector (FXRD-4343FAW, VIEWORKS, 19 Anyang, Korea) composed of a thin-film transistor based on amorphous silicon with a flexible plastic substrate. They performed a quantitative evaluation via experiment to justify its viability where the modulation transfer function, noise power spectrum, and detective quantum efficiency were investigated for the proposed and comparative detectors. The authors justified that novel detector can efficiently increase the image performance in noise characteristics particularly. Finally, they concluded that the proposed detector confirmed the viability of mobile X-ray devices that require low doses. The novel detector can improve the image quality and work convenience in extended radiography.

Comments:

1  1) Please correct eq. 3 for ???? (?, ?), in opinion of this reviewer, the sum in this equation should be double in l, m indexes.

  2) This reviewer did not understand the purpose of the part of discussion in sect 3. Results and Discussion, presented in lines 302-327. It is known and justified that if the noise performance better, using a method of restoring radiographic image by deconvolving the PSF can improve image quality; the level of noise intensity limits the deconvolution quality. The authors mentioned the protocol proposed in their paper [25], maybe it would be interesting to present as one concrete example the usage of the mentioned protocol for their novel detector?

Author Response

Thank you for review and comment in this manuscript.
We have revised the paper as your suggestion and responded point by point.
Please confirm attached revised manuscript and response files.

Best regards,

Youngjin Lee

Reviewer 2 Report

Review of manuscript Diagnostics # 1881374

Characterization of Flexible Amorphous Silicon Thin-Film Transistor-Based Detectors with Positive-Intrinsic-Negative Diode in Radiography

Bongju Han, Minji Park, Kyuseok Kim, and Youngjin Lee

Summary

The aim of this study is to compare and quantify the performances of three a-Si TFT-based detectors with a flexible plastic substrate. Interest of such technology is to perform radiography under “ALARA” radiation exposure conditions. Two parameters were studied: the pixel size and the thickness of the scintillator part. Measurements show that the detector with the highest scintillator thickness and the smallest pixel size has the better performances.

Main comments

I am a physicist and instrumentalist working on a spectroscopy synchrotron beamline, that is to say I am not familiar with all the aspects of this work.

My main comment concerns the choice of the detectors, all from the same company from which the first author is affiliated. For me there is a potential conflict of interest which has to be indicated. To avoid this impression, comparisons of the performances with other detectors from other companies, would have been useful, even using measurements from the literature as the performances were quantified using standardized protocols.

The description of the used detectors has to be improved. For example, a schematic cross-section of the detection area should be interested to better understand the different parts of it. The thickness of the scintillator part is claimed to increase from “low” to “high”. It is quite short for a description, and real values have to be indicated. On the same subject and in the conclusion, it is said that “The proposed FXRD-4343FAW detector is characterized by a certain level of spatial resolution”: “certain level of spatial resolution” is not really precise to my opinion.

The total size of the detection area is not indicated in table 1.

Image performances were quantified following well-described protocols.

Spatial resolution of the detectors was quantified by the modulation transfer function. From this point of view the performance of the FXRD-4343FAW detector is lower (or similar depending of the error-bare of the measurement) than that of the FXRD-4343VAW Plus one. This point has to be more clearly mentioned in the conclusion.

Concerning the contrast to noise ratio and the coefficient of variation, the FXRD-4343FAW has improved performances compare to the two others.

Details:

In the abstract: “This scintillator has a thallium-doped cesium iodide scintillator with…”: “This detector”?

Author Response

(The authors gave the same response as above.)
